# Comparison of Amblyopia Treatment Effect with Dichoptic Method Using Polarizing Film and Occlusion Therapy Using an Eye Patch

**DOI:** 10.3390/children9091285

**Published:** 2022-08-26

**Authors:** Yo Iwata, Tomoya Handa, Hitoshi Ishikawa

**Affiliations:** 1Department of Rehabilitation, Orthoptics and Visual Science Course, School of Allied Health Sciences, Kitasato University, 1-15-1 Kitasato, Sagamihara 252-0373, Japan; 2Department of Ophthalmology, School of Medicine, Kitasato University, 1-15-1 Kitasato, Sagamihara 252-0374, Japan

**Keywords:** amblyopia, anisometropia, dichoptic treatment, polarizing film

## Abstract

We developed a novel, low-cost, easily administered method that uses a polarizing film to enable dichoptic treatment for amblyopia. In this study, we compared its effects with occlusion therapy using an eye patch. Fifty-eight patients (aged 4.7 ± 1.0 years) diagnosed with anisometric amblyopia were included and instructed to wear complete refractive correction glasses with either occlusion therapy using an eye patch (eye patch group) or dichoptic treatment using polarizing film (polarizing film group) for 2 h per day. We examined the improvement in the visual acuity and compliance rate of the patients 2 months after treatment initiation. After treatment, the polarizing film group showed significant improvement in visual acuity compared with the eye patch group. Moreover, the compliance rate was significantly better in the polarizing film group than in the eye patch group. In both groups, there was a significant correlation between the improvement in visual acuity and compliance rate. This new dichoptic treatment using a polarizing film was shown to be effective for anisometropic amblyopia.

## 1. Introduction

Amblyopia is a condition in which visual acuity remains poor despite refraction correction and the absence of organic disease. Its prevalence varies across geographical locations and has been reported to be approximately 1–2% [1,2]. Furthermore, amblyopia is a common cause of visual impairment in children and occurs approximately 10 times more frequently than other eye diseases such as retinopathy of prematurity, optic nerve atrophy, and nystagmus in childhood [3].

The first choice for amblyopia treatment is the wearing of eyeglasses for complete refraction correction; however, it has been reported that wearing spectacles only helps among approximately 30% of patients [4]. Therefore, occlusion therapy using an eye patch is most commonly used as an additional treatment [5]. It aims to improve visual acuity through the forcible use of the amblyopic eye by blocking the sound eye. However, occlusion therapy has side effects including occlusion amblyopia, mental distress, and skin irritation [6,7]. Given that occlusion therapy has low compliance, it is administered only for approximately 50% of the prescribed occlusion time [8,9]. Furthermore, it has been reported that compliance decreases with an increase in the treatment period [10]. In contrast, in addition to the eye-patch-based occlusion therapy, perceptual learning paradigms that improve visual acuity by permanently improving the performance on perceptual tasks as a result of experience and practice [11,12,13], and binocular open amblyopia treatment devices that aim to improve visual acuity by presenting separate visual stimuli to the sound eye and the amblyopic eye have been developed in recent years [14,15,16,17,18]. In binocular open amblyopia treatment devices, visual stimuli are presented separately to the sound and amblyopic eyes. It is known that in patients with amblyopia, the difference in the visual acuity of the amblyopic and sound eyes increases with suppression, which worsens stereoscopic vision [19]. Devices for dichoptic treatment for amblyopia reduce interocular suppression and aim to improve visual acuity and stereoscopic vision through tasks such as games by unifying the input balance from each eye and enabling the integration of visual information from each eye [20]. Furthermore, amblyopia is caused by suppression in the visual cortex, which occurs with binocular opening [21]. Under the condition of one eye occlusion with the eye patch, amblyopia treatment is performed in a deviation from the condition of binocular opening, where suppression occurs. On the other hand, amblyopia treatment under binocular opening may be more effective for suppression removal because it maintains binocular opening and provides visual stimulation to the amblyopic eye. Several types of dichoptic treatment devices have been developed for amblyopia including digital devices such as tablet terminals, smartphones, and head-mounted displays [14,15,16,17,18]. The environment for amblyopia treatment is limited because these devices require specialized equipment, processing, and specialized applications. Furthermore, since these electronic devices are expensive, they are difficult to procure compared to an eye patch, which creates the need to develop a more affordable and easy-to-use dichoptic treatment device for amblyopia [22].

We have therefore developed a dichoptic amblyopia treatment device using polarizing films to achieve this [23]. However, the treatment effect of this method has not yet been compared to other treatment methods. Therefore, in this study, we compared the effectiveness of amblyopia treatment between the dichoptic treatment using polarizing film and one eye occluded using eye patches.

## 2. Materials and Methods

We developed a new dichoptic treatment method for amblyopia using polarizing films. When two sheets of polarizing film are stacked to create a perpendicular absorption axis, only the overlapping portion of the polarizing film prevents light from passing through and occludes the vision (Figure 1a). When the axis is at an angle, light leakage occurs (Figure 1b). One sheet of polarizing film was attached to the target used for the treatment; the target was not limited to electronic devices such as televisions and tablet terminals, but also included paper media such as picture books. Using a clip, another polarizing film with the absorption axis rotated 90° was attached to the treatment glasses on the side of the patient’s sound eye (Figure 2). As a result, the sound eye with a polarizing film attached, only the part of the TV, tablet, book, etc. that has the polarizing film attached, will appear black, while all other parts will appear as normal (Figure 3). The amblyopic eye can see the area where the polarizing film is attached, and all other areas are also visible as normal. This implies that this method maintains a peripheral fusion stimulus that is not present when the eye patch is used. For example, when operating a tablet device, the screen of the device appears black to the sound eye, whereas the body part of the device, the fingers operating it, and all other areas are visible as normal. The amblyopic eye can see everything as normal including the screen of the tablet device. Therefore, this method made dichoptic treatment for amblyopia easy and affordable without limiting the targets that could be used. For each patient, one polarizing film attached to eyeglasses and five polarizing films of different sizes attached to the target were provided (60 × 106 cm, 40 × 71 cm, 30 × 53 cm, 20 × 36 cm, and 7 × 12 cm). If the size of the polarizing film did not fit, the patient was instructed to cut and attach the polarizing film according to the fit. The polarizing film was marked with the directions (back and front, top and bottom) for attaching it.

The study included patients diagnosed with anisometropic amblyopia who visited our hospital between April 2019 and April 2021. The following selection criteria were applied: patients aged ≥3 and ≤8 years with anisometropia of at least 2.00 diopter at equivalent spherical power under accommodation paralysis and a maximum visual acuity of a logMAR value of ≤0.1 in the affected eye. The exclusion criteria were as follows: astigmatism of at least 1.50 diopter, heterophoria of at least 15Δ, strabismus, a history of amblyopia treatment, and difficulty in performing the examination.

Treatment for amblyopia using eye patches (eye patch group) was performed on 34 of the 58 participants (4.7 ± 1.0 years, 3–7 years). In addition, dichoptic treatment for amblyopia using polarizing films (polarizing film group) was performed in 24 cases (Figure 4). Eye patches or polarizing films were provided free of charge to the patients. The numbers between the two groups were not equal because the patient’s guardians were allowed to decide the choice of treatment.

All patients were instructed to use fully refraction-correcting spectacles prescribed in the cycloplegia using cyclopentolate hydrochloride at all times. The eye-patch group was instructed to undergo occlusion therapy using an eye patch for 2 h per day. During amblyopia treatment, the children were instructed to perform activities that involve visual tasks such as watching TV, operating tablets and smartphones, reading books, and other plays. The polarizing film group was instructed to undergo dichoptic treatment for amblyopia using polarizing films for 2 h per day. During the treatment for amblyopia, the patients were instructed to watch TV, use tablet terminals and smartphones, and read books with a polarizing film. All patients were instructed to record the duration (minutes) of treatment for amblyopia performed per day (compliance). We compared the visual acuity improvement 2 months after the start of treatment for amblyopia, the compliance rate [(treatment implementation time/treatment instruction time) × 100] for the amblyopia treatment, and the correlation between the visual acuity improvement and compliance rate.

The Mann−Whitney U test was used to compare the eye-patch and polarizing film groups. The Bayes factor (BF) was calculated for the patient age, anisometropia, and before treatment visual acuity. The effect size was calculated to compare the improvement in visual acuity between the two groups. The Kendall rank correlation coefficient was used to correlate the visual acuity improvement values with the compliance rate. The normality of the data was confirmed using the Kolmogorov–Smirnov test. Statistical significance was set at *p* < 0.05.

## 3. Results

We confirmed that all patients were able to wear the glasses at all times. No significant differences were noted in patient age, anisometropia, and visual acuity between the eye-patch and polarizing film groups (*p* = 0.26, *p* = 0.95, *p* = 0.18) (BF = 1.5, BF = 2.8, BF = 1.1) before treatment for amblyopia. The mean age, anisometropia, and amblyopic visual acuity were 4.8 ± 1.0 years, 3.24 ± 0.66 diopter, and 0.26 ± 0.10, respectively, in the eye patch group before treatment. The mean age, anisometropia, and amblyopic visual acuity were 4.5 ± 1.0 years, 3.23 ± 0.64 diopter, and 0.30 ± 0.11, respectively, in the polarizing film group before treatment (Table 1).

This study was approved by the Ethics Review Board of Kitasato University School of Medicine/Hospital and was performed in accordance with the tenets of the Declaration of Helsinki. Informed consent was obtained from all participants and a legal guardian. The participants underwent either occlusion therapy using an eye patch or dichoptic treatment for amblyopia using polarizing films and were free to choose between the two methods.

The visual acuity of the eye patch and polarizing film groups after treatment for amblyopia was 0.17 ± 0.10 and 0.13 ± 0.10, respectively, with a visual acuity improvement of −0.08 ± 0.07 and −0.17 ± 0.07, respectively. The polarizing film group showed significant improvement in visual acuity compared to that in the eye-patch group (*p* < 0.0001, r = 0.578). The compliance rates of the eye patch and polarizing film groups were 68.8% ± 18.8% and 85.3% ± 15.4%, respectively (Table 1), with significantly better values in the polarizing film group (*p* = 0.002). A significant correlation was found between the improvement in visual acuity and compliance rates in both groups (eye-patch group, *p* = 0.003, r = −0.47; polarizing film group, *p* = 0.002, r = −0.67) (Figure 5).

## 4. Discussion

In this study, we administered dichoptic treatment using polarizing films to patients with amblyopia, and its effectiveness was examined. This showed a significant improvement in visual acuity compared with occlusion therapy for amblyopia using an eye patch. Dichoptic treatment for amblyopia is expected to have a better effect than occlusion therapy for amblyopia, and several studies have reported its superiority [14,24,25,26,27]. However, some studies have reported no difference between the two methods, making it difficult to reach a definite conclusion [15,28,29]. Furthermore, it has been reported that dichoptic treatment does not completely block the sound eye like the eye patch, which decreases mental burden and maintains compliance [14,16]. In the present study, superior compliance to dichoptic treatment was observed. The difference between the sound eye and the amblyopic eye in the amblyopic treatment with both eyes open in this study was that the sound eye sees the part to which the polarizing film was attached as black, while in the amblyopic eye, that part was visible. In addition, all other parts were visible to both eyes. This was the same as the conventional amblyopia treatment device under binocular opening, and the treatment was performed while binocular vision was maintained. However, the previously reported binocular amblyopia treatment devices that used anaglyphs such as the dichoptic Tetris game and Dig Rush [15,30] separate the vision of the sound eye from that of the amblyopic eye on the screen of the tablet device. This was somewhat different to the method using polarizing film in this study. Therefore, there may be a difference in the effectiveness of the treatment due to this difference in visibility.

We were unable to determine whether the superior visual acuity of this method compared to those treated with the eye patch was a result of the differences in the principles of dichoptic and occlusion therapy or the difference in compliance with the treatment. Although the improvement in visual acuity was 2.1 times higher in the polarizing film group than in the eye-patch group, the difference in the compliance rate remained at 1.2 times (85.3/68.8). These results suggest that dichoptic treatment may be more effective than occlusion therapy. However, since the compliance in this study is based on self-reported data by the patient, it cannot be denied that the reported values may deviate from the actual values. It has been pointed out that the compliance values reported by patients deviate from the actual values [10]. The same was true for eyeglass compliance. We confirmed that all patients were able to wear their glasses at all times; however, since this was self-reported, we could not be certain about this.

Several dichoptic treatment methods for amblyopia have been developed thus far [14,15,16,17,18]; however, all devices require specialized devices and applications. Therefore, visual stimuli used for amblyopia treatment are limited and expensive, making it difficult to provide these to all patients, unlike eye patches. However, the method enables dichoptic treatment for amblyopia even on digital devices such as TVs, tablet terminals, smartphones, and paper media such as books. Moreover, since the polarizing film is inexpensive, it can be easily provided to all patients. Therefore, while maintaining a high compliance rate, the binocular open amblyopia treatment can be easily performed in clinics that do not have special equipment.

In this study, by making the axes of the polarizing films orthogonal to each other, only black was visible to the sound eye in the area to which the polarizing films were attached. However, it may be possible to adjust the contrast gradually by shifting the axes of the polarizing films to 80° or 70° instead of orthogonal. Since there are reports that it is crucial to adjust the input of the sound and amblyopic eyes in amblyopia treatment under binocular opening [31], amblyopia treatment with contrast adjustment according to the visual acuity of the amblyopic eye should be studied in the future.

One limitation of this study was that the patients themselves chose the treatment method; hence, it is possible that the group that chose the new treatment method was more aggressive toward the treatment, and this bias could not be eliminated. Additionally, information bias has not been eliminated due to the lack of masked inspectors. In the future, it is necessary to conduct a blinded randomized controlled trial using this study as a reference and to examine in more detail the amblyopia treatment method using polarizing films under open binocular conditions.

## 5. Conclusions

This new dichoptic treatment using a polarizing film was shown to be effective for anisometropic amblyopia. This method can be easily performed in clinics that do not have special equipment.

## Figures and Tables

**Figure 1 children-09-01285-f001:**
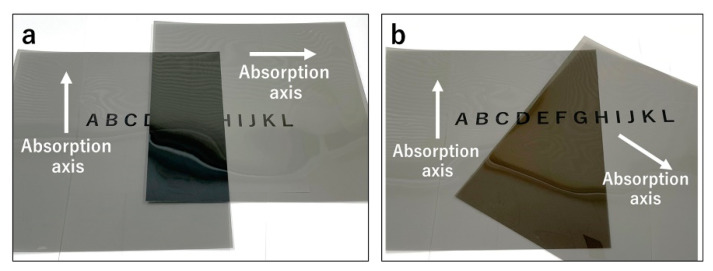
(**a**) The underlying principle of the dichoptic treatment method applied in this study. Light is unable to pass through two polarizing films when their absorption axes overlap so that they are perpendicular to each other. (**b**) If the absorption axis of the polarizing film is not perpendicular, light leakage will occur.

**Figure 2 children-09-01285-f002:**
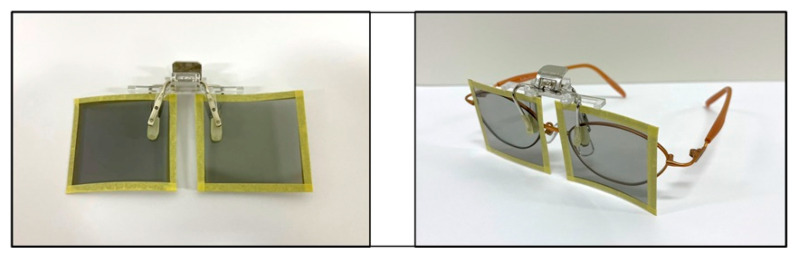
The polarizing film attached to the eyeglasses. The film can be easily attached to the eyeglasses using a clip.

**Figure 3 children-09-01285-f003:**
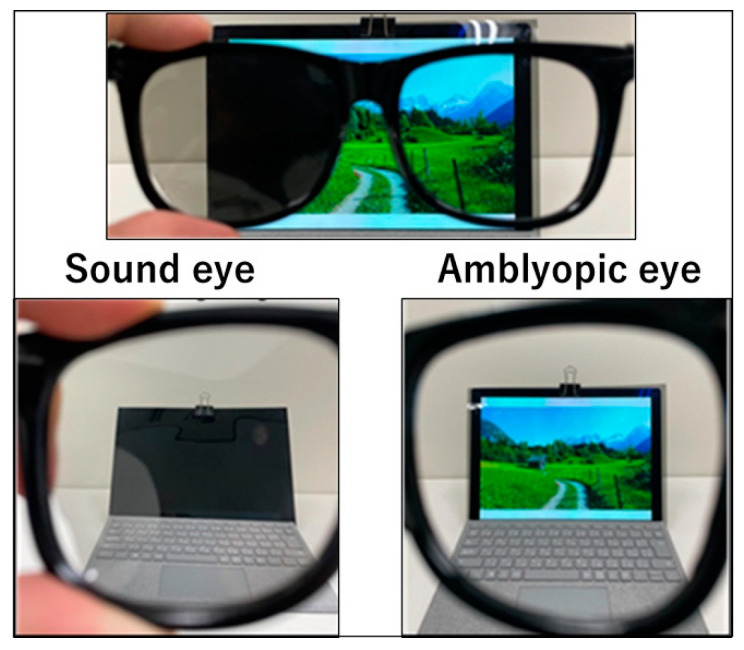
The dichoptic treatment method used in the study. A polarizing film was attached to the monitor for amblyopia treatment, and another polarizing film was attached to the eyeglasses on the sound eye side so that the absorption axes of the films were perpendicular to each other. No polarizing film was attached to the side of the eye with amblyopia. Thus, only the eye with amblyopia could see the monitor when both eyes were open.

**Figure 4 children-09-01285-f004:**
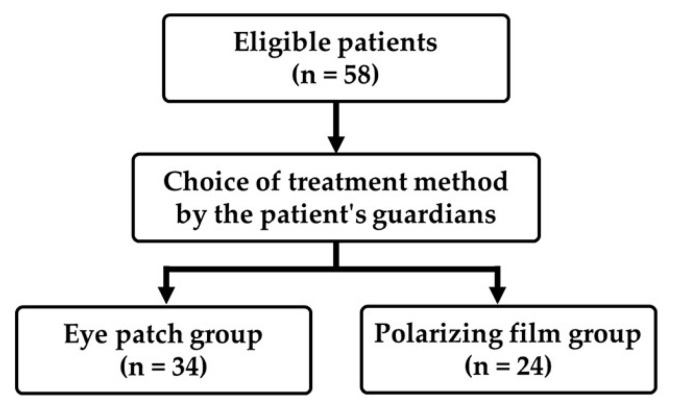
The flow diagram of this study. The treatment method was determined by the choice of the patient’s guardian.

**Figure 5 children-09-01285-f005:**
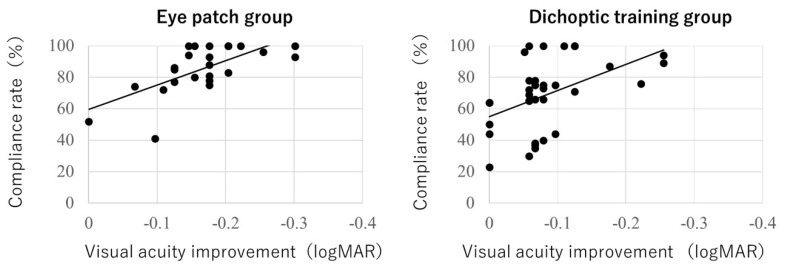
The correlation between the visual acuity improvement and compliance rate. A significant correlation was observed between the visual acuity improvement and the compliance rate in both study groups.

**Table 1 children-09-01285-t001:** The age, anisometropia, visual acuity before and after treatment, value of improvement in visual acuity with treatment, and compliance rate in this study are shown here. Significant differences were found in the visual acuity improvement values and compliance rates.

	Age (Years)	Anisometropia (Diopter)	Visual Acuity before Treatment	Visual Acuity after Treatment	Visual Acuity Improvement	Adherence Rate (%)
Eye patch group	4.8 ± 1.0	3.24 ± 0.66	0.26 ± 0.10	0.17 ± 0.10	−0.08 ± 0.07	68.8 ± 18.8
Polarizing film group	4.5 ± 1.0	3.23 ± 0.46	0.30 ± 0.11	0.13 ± 0.10	−0.17 ± 0.07	85.3 ± 15.4

## Data Availability

All data used to support the findings of this study are available from the corresponding author upon request.

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
