# Peer review of "Comparison of Amblyopia Treatment Effect with Dichoptic Method Using Polarizing Film and Occlusion Therapy Using an Eye Patch"

_children, 2022, doi:10.3390/children9091285_

Round 1

Reviewer 1 Report

Overall the manuscript is well written and conveys the particular meaning of Your research 

There are a few corrections which should be carried out :

line 27: It is recommended to change the definition of amblyopia deleting the term “disease”

line 80: it is recommended to highlight that using “polarizing film” compared to “eye patch” the peripheral binocular vision stimulus remains

line 114-116: It is recommended to move these lines in the results section at line 146

line 114-116: it would be useful to explain why the two patient groups did not have the same number of participants

line 119: it is recommended to change “for accommodation paralysis with” with “in cycloplegia using”

Reviewer 2 Report

In my opinion, it is an interesting paper, which provides new knowledge and also proposes a simple and economical therapy.

It is well developed and explained so, I do not recommend changes.

Reviewer 3 Report

General comments

The manuscript is well written and structured and presents relevant information that may assist practitioners in the choice of treatment for amblyopia.

I offer some comments and suggestions to the authors for their consideration.

Introduction

The introduction would benefit from hypotheses providing a rationale.

Materials and Methods

Sample size calculation details need to be provided.

Add a flow diagram to population and their distribution.

Results

You must include a table that provides a description of the experimental results and their interpretation.

Discussion

I would like to be able to distinguish a first paragraph with a summary of what was sought and what was found in this work, a discussion of the relevant findings together with what was found in other works, an explanation of the meaning of these findings and explanation of the implications for practice clinic.
